# Impact of driving characteristic parameters and vehicle type on fuel consumption and emissions performance over real driving cycles

**Elmira Bagheri**[1], **Masoud Masih Tehrani**[1], **Mohammad Azadi**[2]*, **Ashkan Moosavian**[3]

**1** School of Automotive Engineering, Iran University of Science and Technology, Tehran, Iran, **2** Faculty of Mechanical Engineering, Semnan University, Semnan, Iran, **3** Department of Agricultural Engineering, Technical and Vocational University, Tehran, Iran

* m_azadi@semnan.ac.ir

## Abstract

With the growing need for sustainable transportation solutions, understanding the relationship between driving characteristic parameters, vehicle type, and their impact on emissions and fuel consumption over real driving scenarios is becoming increasingly important. In this paper, four conventional vehicles and one hybrid vehicle with different technologies were compared in four distinct routes in Tehran city. Nineteen real driving cycles were generated using widely employed K-means and PCA algorithms. The vehicles were simulated on MATLAB/Simulink according to their specifications. Twelve driving characteristic parameters, fuel consumption, CO, NOx, HC, and $CO_2$ of vehicles with different powertrains, engines, and body styles were calculated over real and standard driving cycles. Notable findings show that driving characteristic parameters exhibit distinct influences on fuel consumption and emissions, depending on the specific driving conditions and vehicle type. Additionally, the hybrid vehicle achieved 39% and 26% fuel savings compared to gasoline and dual fuel vehicles, respectively. However, it emitted significantly higher levels of CO and HC. In contrast, the turbocharged vehicle increased CO and HC emissions compared to the naturally aspirated vehicle, but consumed less fuel (approximately 6%) and emitted lower amounts of $CO_2$ (approximately 19%). In real driving cycles, the sedan vehicle generally exhibited slightly lower values compared to petrol SUV due to lower weight and drag coefficient.

## Introduction

In today's world, where the increasing demand for energy resources and the reduction of fossil fuels are pressing concerns, it is imperative to prioritize environmental preservation, minimize air pollution, operate within fuel supply constraints, optimize fuel usage, and enhance engine performance [1]. Implementing Intelligent transportation systems, such as electric vehicles, car-sharing programs, and intelligent traffic management, can significantly contribute to these

**Data Availability Statement:** All codes related to this study (for the vehicle simulation with Simulink and for the driving cycle with Matlab) are publicly

available at: https://data.mendeley.com/datasets/
f7c75syg2f/1 (Azadi, Mohammad; Bagheri, Elmira;
Masih-Tehrani, Masoud; Moosavian, Ashkan
(2024), "Codes for the vehicle simulation with
Simulink and for the driving cycle with Matlab",
Mendeley Data, V1, doi: 10.17632/f7c75syg2f.1).

**Funding:** The author(s) received no specific
funding for this work.

**Competing interests:** The authors have declared
that no competing interests exist.

goals by reducing emissions and conserving energy [2, 3]. On the other hand, amount of fuel consumed and the level of emissions are closely linked to the specific driving habits observed in different geographical areas. One common method of illustrating these driving habits in a particular region is by using driving cycles [4]. It is essential to thoroughly examine these cycles, as they exhibit notable variations from one region to another.

Typically, driving cycles are employed to evaluate the performance of vehicles, including emission rates and fuel consumption [5], as well as in vehicle simulations [6]. In recent years, numerous studies have been conducted to decrease fuel consumption and emissions. These studies can be analyzed from different perspectives and can be categorized according to their common features, including cleaner internal combustion engines, hybrid or electric vehicles [7], engine downsizing, alternative fuels, and improving the road network topography [8]. However, one crucial and often overlooked factor is the behavior of the driver, which significantly impacts a vehicle's fuel consumption and emissions. Researchers have observed that driving habits and energy efficiency of vehicles are influenced by traffic conditions [9, 10]. Their findings suggest that in congested areas, average fuel consumption tends to increase, regardless of the specific road section [11]. Furthermore, studying changes in driving characteristic parameters, such as instantaneous acceleration and speed, is essential because they have a significant effect on exhaust emissions and fuel consumption [12]. Recent advancements in methodologies, particularly the application of Geographic Information Systems (GIS), have provided valuable insights into the complexities of driving behavior [13]. For instance, several studies have implemented GIS to georeference naturalistic driving data, enhancing data representation and visualization, thus enabling researchers to better understand driving patterns [14]. Moreover, innovative frameworks utilizing GIS allow for the extraction and analysis of driving behaviors across various contexts and spatial scales, ultimately contributing to a more comprehensive evaluation of how driver behaviors influence performance metrics such as fuel consumption and emissions [15].

Chong et al. [16], analyzed vehicles using a Portable Emissions Measurement System (PEMS) that included a Global Positioning System (GPS). They found that higher levels of NOx emissions were associated with increased speeds, stronger accelerations, and reduced exhaust gas recirculation. During engine conversions, excessive emissions of HC and CO were frequently observed, particularly during acceleration. Sofwan and Latif [17], found that operating vehicles at low speeds, which improves fuel economy, leads to reduced emissions of CO and NOx. The results were obtained through on-board emissions testing and were conducted on main streets in the heart of Kuala Lumpur's urban environment. However, this operating condition is associated with increased emissions of $CO_2$. Relying solely on average speed does not adequately capture driving behavior.

The type of vehicle is another factor that can influence changes in emissions and fuel consumption. Wang et al. [18], investigated the fuel consumption and emissions of Hybrid Electric Vehicles (HEVs) and conventional vehicles using Worldwide Harmonized Light Vehicle Test Cycles (WLTC) and real driving emission tests. One of the conventional vehicles was turbocharged, and the results showed that the fuel consumption was similar for both types of vehicles, but HEVs had higher CO emissions, while conventional vehicles had higher NOx emissions. The study also found that the behavior of naturally aspirated and turbocharged vehicles differed on various routes. Huang et al. [19], compared the real-world fuel consumption and emissions of conventional vehicles and HEVs. They found that both types of vehicles consumed more fuel in real driving conditions, and HEVs had higher HC and CO emissions than conventional vehicles. Wang et al. [20], examined how much emissions could be reduced by using HEVs in comparison to conventional vehicles, using real driving cycles from Metropolitan Beijing and Toronto. The results showed substantial reductions in $CO_2$ and NOx

emissions when using HEVs, particularly when the vehicles were operating at low power demand in Beijing. However, in Toronto, the benefits of HEVs were minor due to more aggressive driving patterns.

Pignatta and Balazadeh [21], assessed the emissions of six conventional vehicles, including five sedans and one Sport Utility Vehicle (SUV), and two HEVs in real driving conditions in Iran. The findings indicated that HEVs demonstrated lower fuel consumption and relatively decreased exhaust emissions compared to conventional vehicles. In conventional vehicles, NOx emissions increased with vehicle speed, whereas hybrid vehicles did not exhibit a decisive trend between NOx emissions and vehicle speed. Additionally, the SUV had greater exhaust emissions and fuel consumption compared to sedans. Baêta et al. [22], studied the fuel consumption and emissions of a turbocharged engine. They found that combining direct injection and turbocharging improved thermal efficiency and reduced emissions of HC and NOx.

One limitation of previous research is that they often focus on a limited number of factors, neglecting others when studying the impact on a particular type of vehicle. In other words, it is challenging to find a study that has thoroughly examined all the factors that influence fuel consumption and emissions in various vehicles at the same time and compared them to each other. Furthermore, comparisons of different powertrains, engines, body styles, and driving characteristic parameters in real driving cycles are some other examples. The wide range of vehicle types and their distinct characteristics make it impractical to adopt a single universal driving cycle for all of them, which can result in either underestimating or overestimating emission factors. In contrast to previous studies that relied on standardized drive cycles developed in America and Europe or real data of fuel consumption and emissions, our study creates real driving cycles that are specifically tailored to the unique driving patterns of emerging automobile markets like Iran. This is achieved by collecting data along four different route types and using a combination of Principal Component Analysis (PCA) and K-means clustering to develop driving cycles that accurately represent the driving patterns of these regions.

The innovations of this paper can be summarized as follows:

- Development of real driving cycles using K-means clustering and PCA, combined with real driving data from four distinct routes and one driver to capture representative driving patterns.

- Utilization of a simulation-based approach with MATLAB Powertrain Blockset and Simulink Design to simulate reference vehicles and generate fuel consumption and emissions data for real driving cycles, as well as two standard driving cycles.

- Comparative analysis is conducted to examine the impact of different powertrains, engines, and body styles on driving characteristic parameters, exhaust emission behavior, and fuel consumption of conventional vehicles and a hybrid vehicle.

The organization of the remaining sections of this paper is as follows: The methodology section provides a detailed description of the methodology, including sample route selection, vehicle selection, data acquisition, data processing techniques, and the development of driving cycle. The vehicle and simulation section elaborates on the simulation of reference vehicles. The results and discussion section examines the effects of driving characteristic parameters, powertrains, engine downsizing, body styles, and their impact on fuel consumption and emissions. Finally, the conclusions and future works section summarizes this study by highlighting the major findings and key outcomes.

## Methodology

In this section, the methodology employed in this study is explained in detail, with a representation of the steps outlined in Fig 1. The subsequent explanation is also provided.

Firstly, the sample routes are calculated by considering traffic data from various types of roads and populous areas in Tehran city. Then, vehicles are selected based on their specifications, such as powertrain, engine types, and body styles, and choose the most popular ones. This ensures that the test vehicles are representative of real driving conditions. To control for individual driving behavior and focus on the impact of vehicle type on driving performance, a single driver drove all sample routes. This allows to isolate the effect of vehicle characteristics on the results. The naturalistic data are then processed to develop driving cycles, which are crucial for understanding the behavior of vehicles under real-world conditions. The 12 driving characteristic parameters calculated are indeed related to kinematic driving behavior, and these parameters are used to identify the most influential factors on the results. By reducing these parameters to two sensitive parameters using PCA, the underlying dynamics of vehicle behavior can better understand.

The K-means algorithm is employed to construct driving cycles for five vehicles in all sample routes. After that, vehicles are simulated in MATLAB/Simulink according to their specifications, and input the related driving cycles as scenarios to each vehicle. By running simulations, fuel consumption and emissions are obtained for each vehicle. This allows to examine the impact of vehicle technologies on fuel consumption and emissions. After analyzing the driving characteristic parameters of each driving cycle, the impact of vehicle technologies along with their corresponding fuel consumption and emissions data, to see how each vehicle affects these parameters.

### Sample route selection, vehicle selection, and data acquisition

Tehran, the capital of Iran and the most populous city in the country, is also the second most populous metropolis in the Middle East. Therefore, dealing with air pollution is one of the most necessary actions in this city. An optimal route for collecting driving data is a stepping stone for developing a drive cycle that can closely represent real driving behavior in a city.

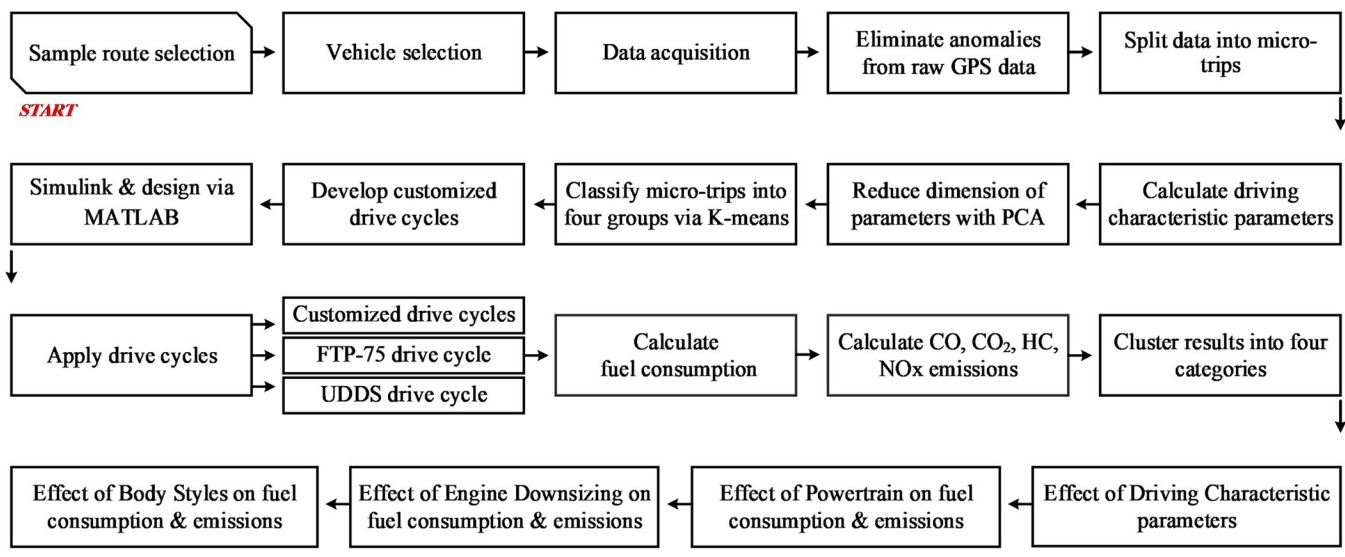

**Fig 1. The steps of the research as a flowchart.**

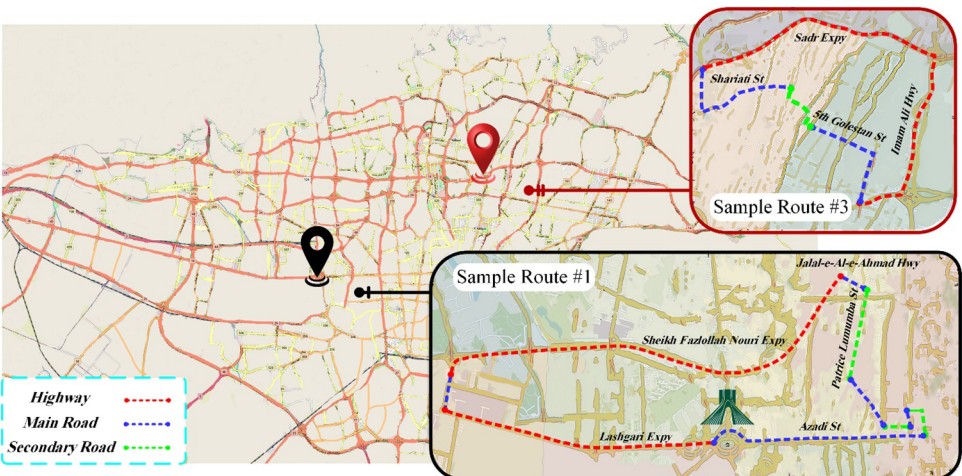

**Fig 2. Representative of sample routes 1 and 3.** Figure created by the authors using Paint.NET, Adobe Photoshop, Microsoft Visio Pro and Autodesk SketchBook.

The sample routes should encompass a diverse range of urban roads [23], including both densely populated and non-populated areas [24]. It is essential to consider the urban topological structure, traffic flow, driving speeds, travel times, and patterns of origin and destination [25]. The sample routes in this study were initiated by analyzing data obtained from the Transport and Traffic Organization (TTO) of previous years in Tehran city. Despite the age of the data, the ongoing relevance of the TTO data lies in the consistent traffic patterns and infrastructure layouts of Tehran city, which have remained relatively stable over time. The report provided statistics on major districts, quantifying the proportion of high-volume traffic in different types of routes in Tehran city. Based on these factors, four circular routes were selected. Fig 2 shows two sample routes, 1 and 3. Additionally, Table 1 provides detailed information about the sample routes, which was derived from GPS data. The remaining two sample routes (2 and 4) were chosen randomly, while maintaining the same starting and ending points as sample routes 1 and 3, respectively. Since sample routes 2 and 4 were selected randomly, their distribution across different types of routes differed from those of sample routes 1 and 3.

In an urban area like Tehran city, the choice of vehicles should adhere to the principle of selecting based on popularity. In selecting vehicles, various factors were considered, including different types of powertrains, the presence of turbocharging, and body styles. To facilitate comparisons of driving characteristic parameters, emissions, and fuel consumption, three vehicle models were selected for each of three powertrains (gasoline, hybrid electric, and dual fuel), two types of engines (turbocharged and naturally aspirated), and two body styles (sedan and SUV).

**Table 1. Specifications of sample routes.**

| Type of roads | Contribution to the data traffic (%) | Contribution to the sample route 1 (%) | Length of sample route 1 (km) | Contribution to the sample route 3 (%) | Length of sample route 3 (km) |
|---|---|---|---|---|---|
| *Highway* | 55.3 | 58 | 16.35 | 57 | 9.1 |
| *Main road* | 34.5 | 32 | 9.00 | 33 | 5.1 |
| *Secondary road* | 11.2 | 10 | 2.80 | 10 | 1.5 |

**Table 2. Technical specifications of test vehicles.**

| Parameter | Symbol | Unit | Vehicle | | | | |
|---|---|---|---|---|---|---|---|
| | | | #1 | #2 | #3 | #4 | #5 |
| Vehicle Name | - | - | Dena | Dena+ | Haima S7 | Pars | Prius |
| Model | - | - | 2020 | 2019 | 2020 | 2021 | 2017 |
| Curb Weight | $m_e$ | Kg | 1258 | 1262 | 1545 | 1165 | 1394 |
| Dimensions | l×w×h | Mm | 4559×1944×1460 | 4559×1944× 1460 | 4498×1830× 1730 | 4498×1704×1410 | 4540×1760×1490 |
| Powertrain Type | - | - | Gasoline | Gasoline | Gasoline | Gasoline /CNG | Gasoline /Electric |
| Engine Type | - | - | Naturally aspirated | Turbocharged | Turbocharged | Naturally aspirated | Naturally aspirated |
| Wheelbase | L | Mm | 2671 | 2671 | 2619 | 2669 | 2700 |
| Max Speed | $u_{emax}$ | km/h | 189 | 205 | 170 | 190 | 180 |
| Motor Peak Power | $p_{emax}$ | Kw | 86 | 112 | 127 | 75 | 90 |
| Motor Peak Torque | $T_{emax}$ | Nm | 155 | 215 | 230 | 153 | 142 |
| Transmission | - | - | 5-Speed manual | 5-Speed manual | 6-Speed automatic | 5-Speed manual | CVT |

In this study, five vehicles are examined, including conventional vehicles #1, #2, #3, and #4, as well as a hybrid vehicle (#5). These vehicles are among the most popular domestic and foreign cars in Iran from 2017 to 2021. The benefits of these vehicles include four-cylinder engines with a volume ranging between 1200 and 1800 cc, which is a common engine configuration in the automotive industry, allowing for generalizability of the results to other vehicles. Table 2 presents the key technical specifications of the five vehicles.

In this research, speed data from vehicles was collected using a GPS logger software, which tracked positions and routes, enabling the monitoring of trips and the retrieval of location, speed, altitude, direction, and various statistics. The software allows for the review of recorded trips at any time and provides output in KML, TXT, and GPX formats. The data collection frequency was set at 1 Hz, a decision made with careful consideration of the trade-offs between data granularity and practicality in handling data. The software was installed on a Samsung Galaxy S21 mobile phone.

The reported data in this study only involves recording data from the vehicles driven by the corresponding author, who was aware of the study's objectives. The driver, a citizen with extensive driving experience of approximately 19 years and familiarity with the road conditions in Tehran, navigated all routes to minimize the impact of their behavior on the test results [26]. It is important to note that the driver's proficiency was consistent across all vehicles used in the study, ensuring a standardized approach to the assessment of factors affecting fuel consumption and emissions, such as gear shift delays, acceleration from vehicle takeoff, and lane changing patterns. This approach allows for a more reliable analysis of vehicle characteristics, ensuring that any observed differences in fuel consumption and emissions can be attributed primarily to the vehicle designs and technologies rather than the driver's behavior. The data collection process took approximately 16 hours and 30 minutes, with a total distance traveled of around 570 kilometers. Data collection was conducted in February, June, and July 2022, under sunny weather conditions to eliminate any potential impact of unfavorable weather. While this approach helps maintain reliable results, it is important to acknowledge that variations in weather can significantly influence driving behavior and vehicle performance. Adverse weather, such as rain or snow, often leads to changes in driving patterns. Also, wet roads can increase drag, requiring more energy (and thus more fuel) to maintain speed. Snow and ice can exacerbate these effects. Furthermore, adverse conditions, particularly cold temperatures or heavy rain, can necessitate greater use of vehicle auxiliary systems like heating, air conditioning, and windshield wipers, which draw additional power, impacting fuel

efficiency and increasing emissions. Additionally, data was not collected during routes or periods of abnormal traffic conditions to ensure consistent results.

## Data processing

After importing the text data files into the primary computational software, the unprocessed data was organized into matrices. The data underwent a consistency check, and any irregularities or inconsistencies were subsequently removed. The data from each trip was then divided into smaller segments known as micro-trips. Each micro-trip was defined by a specific start and end point, where the vehicle is idle (speed of zero km/h) and remains in an idle state until the vehicle returns to an idle state.

In the next step, a set of selected driving characteristic parameters, including percentage of driving, idling, cruising, accelerating, decelerating, average speed of trip, average driving speed, maximum speed, maximum negative/positive acceleration, and the standard deviation of acceleration/speed, were employed to describe the attributes of each micro-trip [27]. This comprehensive selection of parameters is grounded in previous research, highlighting their significance in understanding driving behavior, and their relation with fuel consumption and emissions. The percentage of driving time is calculated from (1), where, $T_{total}$ is the total driving time. $T_{total}$ and $T_{drive}$ are calculated from (2) and (3). Also, the idling time can be calculated from (4).

$$T_{drive}(\%) = \frac{T_{drive}}{T_{total}} \times 100 \tag{1}$$

$$T_{total} = t_2 - t_1 + \sum_{i=2}^{n}(t_i - t_{i-1}) \tag{2}$$

$$T_{drive} = T_{total} - T_{idle} \tag{3}$$

$$T_{idle} = \begin{cases} (t_2 - t_1) : & v_1 = 0 \cap a_1 = 0 \\ 0 : & else \end{cases} + \sum_{i=2}^{n}\begin{cases} (t_i - t_{i-1}) : & v_i = 0 \cap a_i = 0 \\ 0 : & else \end{cases} \tag{4}$$

The percentage of idling time and the percentage of cruising time are obtained from (5) and (6), respectively.

$$T_{idle}(\%) = \frac{T_{idle}}{T_{total}} \times 100 \tag{5}$$

$$T_{cruise}(\%) = \frac{T_{cruise}}{T_{total}} \times 100 \tag{6}$$

The value of $T_{cruise}$ is calculated from (7) and the values of $T_{acceleration}$ and $T_{deceleration}$ are

obtained from (8) and (9), respectively.

$$T_{cruise} = T_{drive} - T_{acceleration} - T_{deceleration} \tag{7}$$

$$T_{acceleration} = \begin{cases} (t_2 - t_1) : & a_1 > acc\_threshold \\ 0 & : & else \end{cases} + \sum\nolimits_{i=2}^{n} \begin{cases} (t_i - t_{i-1}) : & a_i > acc\_threshold \\ 0 & : & else \end{cases} \tag{8}$$

$$T_{deceleration} = \begin{cases} (t_2 - t_1) : & a_1 < -acc\_threshold \\ 0 & : & else \end{cases} + \sum\nolimits_{i=2}^{n} \begin{cases} (t_i - t_{i-1}) : & a_i < -acc\_threshold \\ 0 & : & else \end{cases} \tag{9}$$

The percentage of acceleration time and the percentage of deceleration time are calculated from (10) and (11).

$$T_{acceleration}(\%) = \frac{T_{acceleration}}{T_{total}} \times 100 \tag{10}$$

$$T_{deceleration}(\%) = \frac{T_{deceleration}}{T_{total}} \times 100 \tag{11}$$

Mean driving speed, average travel speed, and maximum driving speed are obtained from (12), (13), and (15), respectively, where the distance value is obtained from (14).

$$V_{d-m} = 3.6 \times \frac{dist}{T_{drive}} \tag{12}$$

$$V_{t-m} = 3.6 \times \frac{dist}{T_{trip}} \tag{13}$$

$$dist = (t_2 - t_1)(\frac{v_1}{3.6}) + \sum\nolimits_{i=2}^{n}(t_i - t_{i-1})(\frac{v_i}{3.6}) \tag{14}$$

$$V_{\max} = Max(V) \tag{15}$$

The average negative acceleration and the average positive acceleration are calculated from (16) and (17), respectively.

$$a_{n\_m} = \left( \sum\nolimits_{i=1}^{n} \begin{cases} 1 : & a_1 < 0 \\ 0 : & else \end{cases} \right)^{-1} \times \sum\nolimits_{i=1}^{n} \begin{cases} a_i : & a_i < 0 \\ 0 : & else \end{cases} \tag{16}$$

$$a_{p\_m} = \left( \sum\nolimits_{i=1}^{n} \begin{cases} 1 : & a_1 > 0 \\ 0 : & else \end{cases} \right)^{-1} \times \sum\nolimits_{i=1}^{n} \begin{cases} a_i : & a_i > 0 \\ 0 : & else \end{cases} \tag{17}$$

Additionally, the deviation from speed and deviation from acceleration are obtained from

(18) and (19), respectively.

$$\bar{v}_{sd} = \sigma_v = \sqrt{\frac{1}{n-1}\sum_{i=1}^{n} v_i^2} \tag{18}$$

$$\bar{a}_{sd} = \sigma_a = \sqrt{\frac{1}{n-1}\sum_{i=1}^{n} a_i^2} \tag{19}$$

Subsequently, the PCA algorithm was applied to reduce the original data to a set of underlying factors, known as principal components. This process involved discarding minor details and retaining the essential information [28]. Each principal component is formed by combining the original variables in a way that captures the most independent information in the data [29]. By employing this approach, new combinations of the original variables referred to $k(x_1, x_2,...,x_K)$, were generated to identify the most significant independent component $k$ denoted as $(PC_1, PC_2,...,PC_p)$.

Each principal component can be represented by the following expressed in (20) [28].

$$PC_K = w_{K1}x_{K1} + w_{K2}x_{K2} + \ldots + w_{KK}x_{KK} \tag{20}$$

Where $PC_K$ denotes the intended principal component, $w_{Kj}$ corresponds to the coefficient assigned to the primary variables, and $x_1$ represents the primary variable.

In the standardization step of PCA, the input variables are normalized to have an average value of zero and a standard deviation of one. This process ensures that the variables are on a comparable scale. The resulting matrix $z$, is derived from (21) [28]. This standardization allows for fair comparison and interpretation of the variables in the subsequent PCA analysis.

$$z_{ij} = \frac{x_{ij} - \bar{x}_j}{s_j}, \ for \ i = \{1, 2, \quad .. \quad , n\} \ and \ j = \{1, 2, \quad .. \quad , q\} \tag{21}$$

In this context, $\bar{x}_j$ corresponds to the data average, with $x_j$ and $s_j$ indicating the standard deviation values. Additionally, calculating the correlation matrix for the initial values offers insights into the level of correlation between each primary variable. Each element, $a_{ij}$, in this matrix illustrates the correlation between variables $i$ and $j$, which is obtained from (22) [28].

$$R = \frac{1}{n}z'z \tag{22}$$

In the computation of eigenvalues ($\lambda$) and eigenvectors related to the correlation matrix, (23) and (24) are derived. By finding solution of (23) and (24), the eigenvalues and eigenvectors are calculated to be equal [28]. Additionally, the resultant special vectors derived from these calculations serve as coefficients for the primary variables in forming the corresponding component. Solving (23), where $I$ is a singular matrix, enables the calculation of the eigenvalues ($\lambda_n$). These eigenvalues represent important characteristics of the correlation matrix and play a key role in subsequent analyses.

$$\det(R - \lambda I) = 0 \tag{23}$$

$$\det(R - \lambda I) = V_h \tag{24}$$

In the initial stage of PCA, the variance for each principal component is calculated using (24). This step provides valuable insights into how each component explains the overall variance within the dataset. Following the extraction of the principal components, the parameters with high variance in these components are identified as sensitive parameters, which are

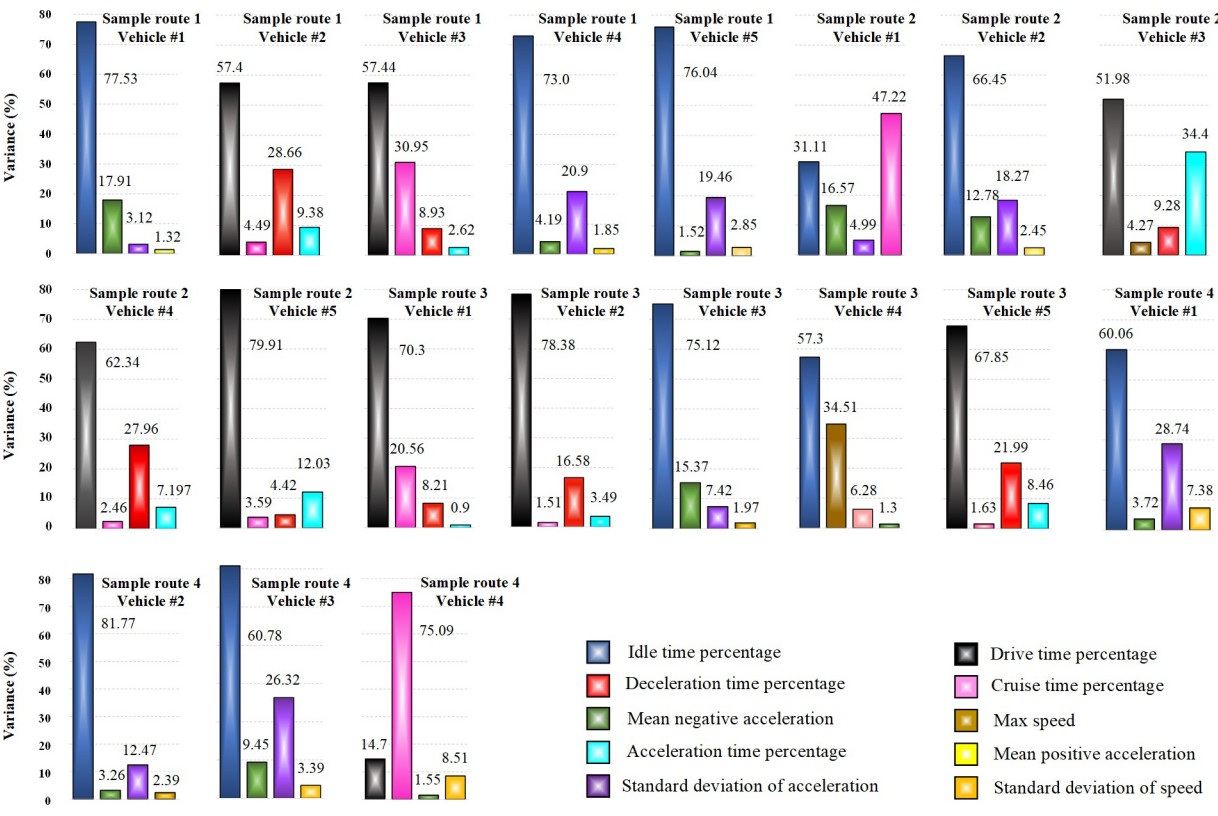

**Fig 3. The variance percentage of driving characteristic parameters.**

deemed to have a significant impact on the underlying patterns and relationships within the data. To ensure a focused analysis, we reduced the dimensionality to the two parameters that accounted for the highest variance percentages. This approach effectively captures the most critical relationships within the data, making these parameters essential inputs for subsequent modeling and analysis. Fig 3 illustrates the distribution of variance percentages for the driving characteristic parameters of different vehicles across various sample routes. The first two parameters of each with the highest variance percentage were considered as sensitive parameters.

## Driving cycle development

In the subsequent step, the mean of sensitive parameters for each micro-trip is utilized in clustering methods to classify the micro-trips into four distinct clusters. Clustering is a statistical method that involves grouping individual samples based on measures of distance, similarity, or variation, while considering important characteristics from the sampled data. Samples with similar values in the representative characteristics are grouped together, and a representative sample is chosen to minimize the collective distance from other group members. In this study, clustering analysis is used to categorize micro-trips with similar conditions, enabling the generation of a chronological order of vehicle speeds that accurately captures the identified driving patterns in the analyzed area. The K-means algorithm is employed as the clustering technique, which is widely used in similar domains due to its efficiency, simplicity, and suitability particularly when dealing with large datasets [30]. The choice of K-means is justified based on several characteristics of the data and the specific objectives of the clustering

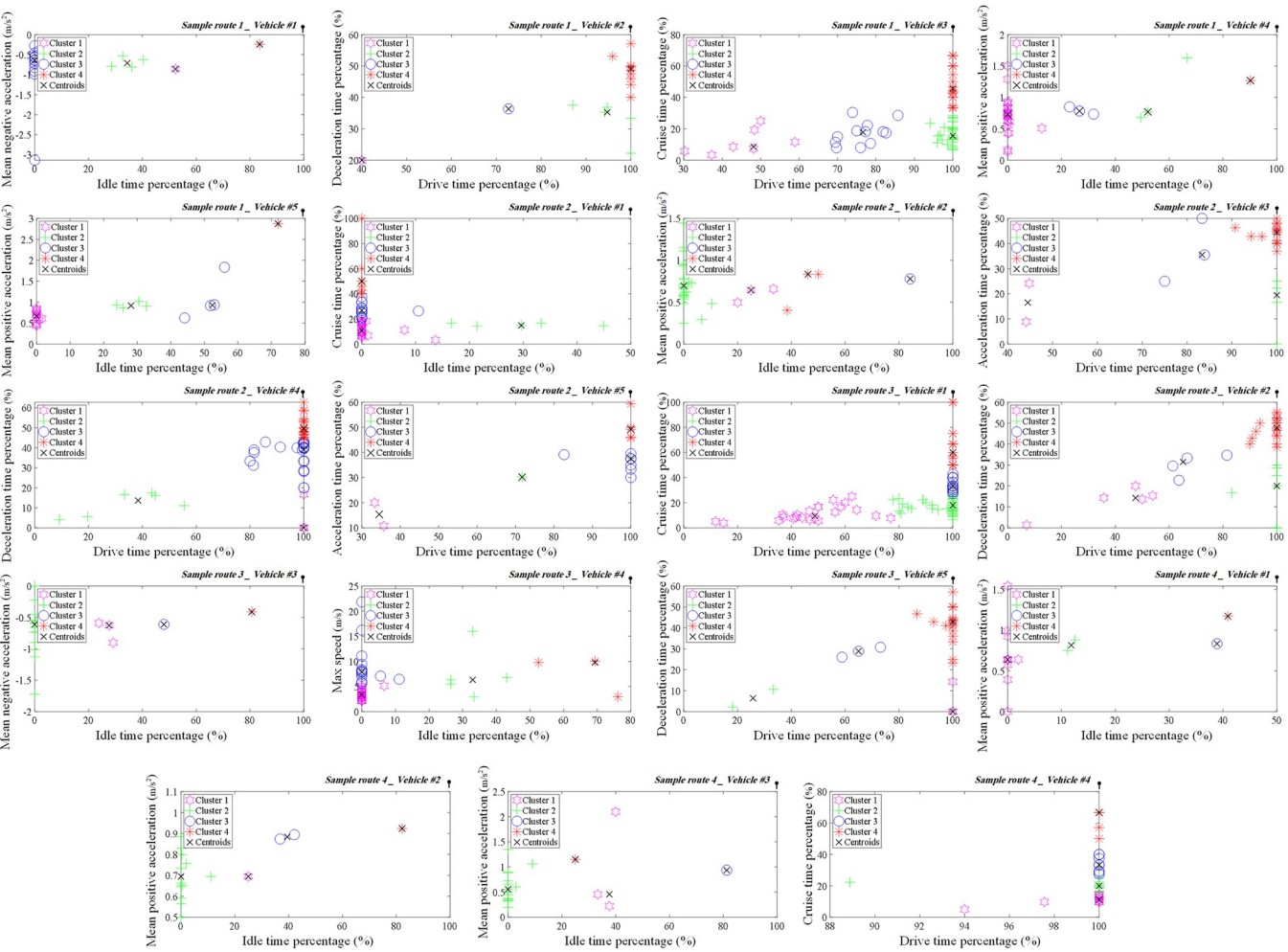

**Fig 4. K-means clustering method based on PCA algorithm.**

approach. The simplicity and interpretability of K-means provide a clear framework for categorizing micro-trips based on distinct driving conditions. Additionally, K-means demonstrates strong scalability, allowing for efficient processing of significant volumes of driving records without compromising performance. The algorithm's distance-based clustering approach, which minimizes intra-cluster variance, is particularly effective for grouping micro-trips that exhibit similar driving behaviors, such as speed patterns. Overall, these factors contribute to K-means being the most appropriate clustering technique for this study. The squared Euclidean distance, as depicted in (25), is the commonly employed measure for evaluating similarity in this approach [31].

$$d_{ij} = \sum_{k=1}^{p} \left( x_{ik} - x_{jk} \right)^2 \tag{25}$$

Each micro-trip consists of $p$ parameters, denoted as $x_{ik}$, where $x_i$ represents the $k$-th parameter of micro-trip $i$, and $d_{ij}$ represents the distance between each pair of micro-trips. Fig 4 illustrates the result of the K-means clustering based on the PCA algorithm applied to all vehicles in the suggested sample routes, which were then used to construct driving cycles.

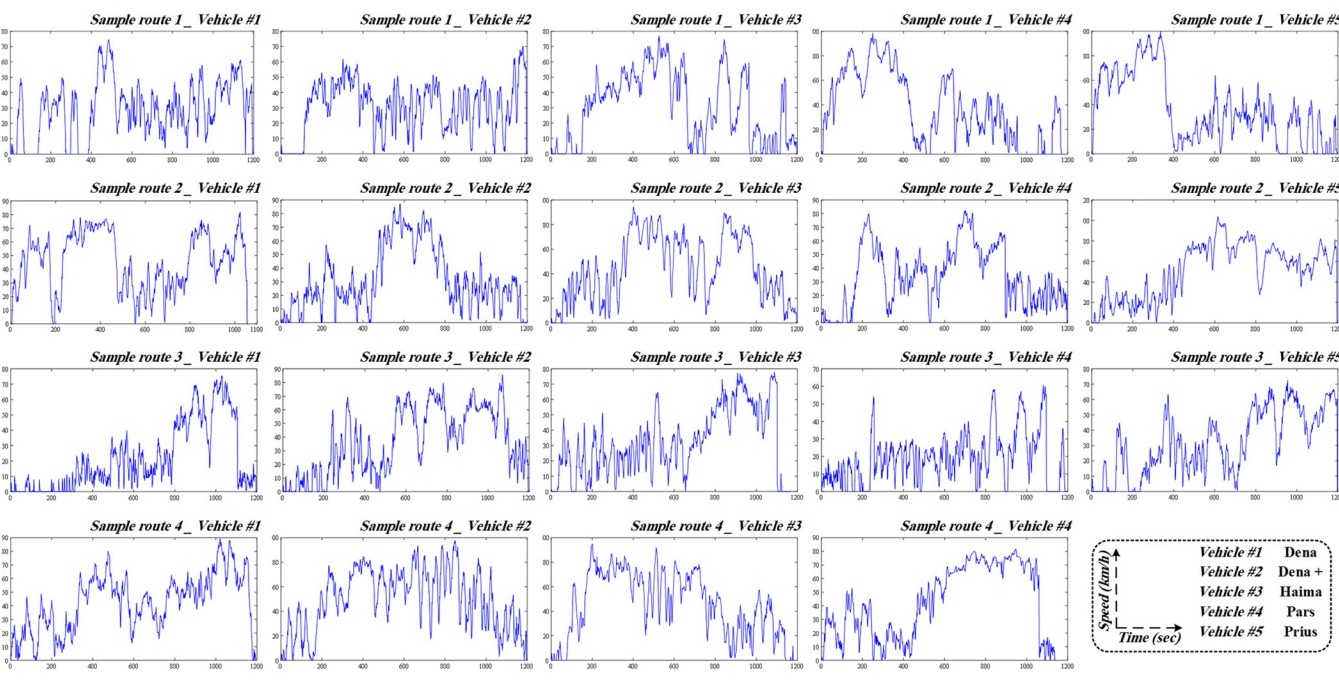

**Fig 5. Developed driving cycles of each vehicle in a distinct sample route.**

Following the application of the PCA algorithm and K-means clustering on micro-trips, it is time to construct driving cycles. The proportion of each cluster in the driving cycle is determined by calculating the time percentage allocated to each cluster. Once the duration of micro-trips in each cluster is determined, micro-trips are selected from each cluster to fill the designated duration of that cluster within the overall driving cycle [32]. This process involves iteratively selecting the closest micro-trips to the center of the cluster until the desired proportion of the cluster is accounted for. As a result, multiple micro-trips from each cluster are chosen to collectively represent the driving cycle. Fig 5 illustrates the driving cycle of vehicles in related sample routes.

## Vehicle simulation

In this section, vehicles are simulated by using MATLAB/Simulink with the Powertrain Blockset, which is a powerful tool for data-driven modeling and analyzing the performance of vehicles [33]. The dynamic modeling of five vehicles and its components is simulated in accordance with the characteristics of each vehicle in this Blockset. In the following, the obtained drive cycles are considered as simulation input and are fed to the model. The Powertrain Blockset provides a comprehensive set of blocks and components that allow to simulation and design of powertrains, including engines, transmissions, drivelines, and other vehicle subsystems. The realistic model of all five vehicles was simulated separately in this environment, each of which consists of six main blocks. The names and connection order of these blocks are shown in Fig 6.

The scenario block is the starting point of modeling, where the behavior of the driver and their maneuvers are modeled in the form of drive cycles. The output of this block serves as input ports for other subsequent blocks. Within the Scenarios block, there are two constant inputs for driver command and driver reference, as well as one feedback input for triggering subsystems. These inputs are defined as 1-D arrays. The concepts of vehicle dynamics are

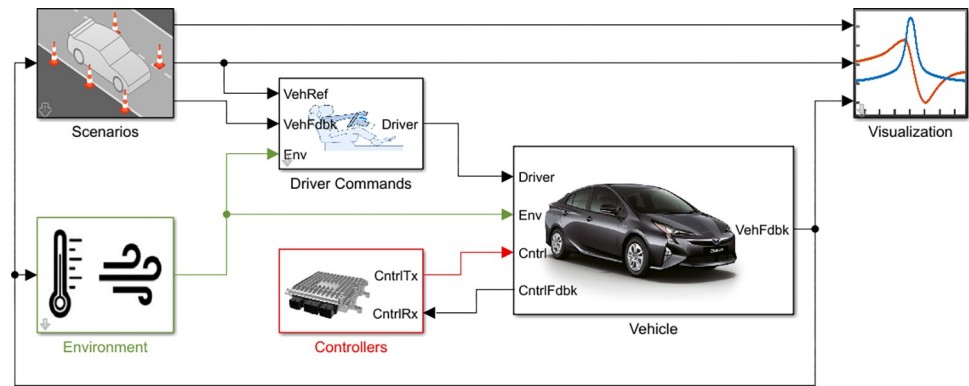

**Fig 6. Steps and structure of designing real vehicle models with MATLAB virtual vehicle composer.**

calculated based on coordinate systems, which are defined in the Z-down and Z-up orientations with the sub-blocks of SAE J670 and ISO 8855, respectively. The parameters of this block are independent of the technical specifications of the vehicle, meaning that the initial values of lateral position and vertical position can be set uniformly for all five vehicles. In this case, the initial values were set to 5.7 and 0.0, respectively.

The driver's behavior and vehicle control are affected by the surrounding environment and road conditions. In order to include these factors, a block named Environment is designed at the beginning of the system and runs parallel to the Scenarios block. The Environment block has a common input port with the scenarios block and its output is connected to other blocks. The parameters of this block include ambient temperature, ambient atmospheric pressure and wind velocity. The wind velocity values in the global inertial frame are represented by three separate ports in X, Y, and Z coordinates, which are connected to each other through an embedded subsystem. In accordance with the detailed analysis of the model, road friction is designed as a subsystem comprising three levels: constant friction, varying friction and three-axle constant friction. These levels are based on wheel parameters, body aerodynamics, suspension system and chassis. The initial values for ambient temperature, ambient atmospheric pressure, and wind velocity for all vehicles are set to 298.15 K (25˚C), 101325 pascal, and 0 m/s, respectively. This provides a consistent set of conditions for a fair comparison of fuel consumption and emissions across all vehicles in this study.

The Driver Commands block can simulate the vehicle's behavior in one of four modes: longitudinal driver, predictive driver, predictive Stanley driver and no driver. In each of these modes, the vehicle reference, vehicle feedback, steering command, gear command, brake command, acceleration command, and environment ports serve as inputs. The primary function of this block is to produce acceleration, braking, gear, and steering signals in various ways. For example, in the longitudinal driver mode, the block generates normalized acceleration and braking commands (between 0 and 1) to track the longitudinal drive cycle by applying a speed-tracking controller. More details of simulating the Driver Command block in Simulink are shown in Fig 7.

In this section, the Driver Commands block with the longitudinal driver mode is used to model the dynamic response of the driver in all vehicles. Additionally, for each signal, three external actions (hold, override and disable) are defined, which are applied to closed-loop commands with a default priority. For each vehicle, three types of rear, neutral and drive shifts (i.e., R, N, D gears) are included with a change time of 0.1 seconds and the velocity gain breakpoints are set between 0 and 100. The details and other parameter values for this block are shown in Table 3 for each vehicle.

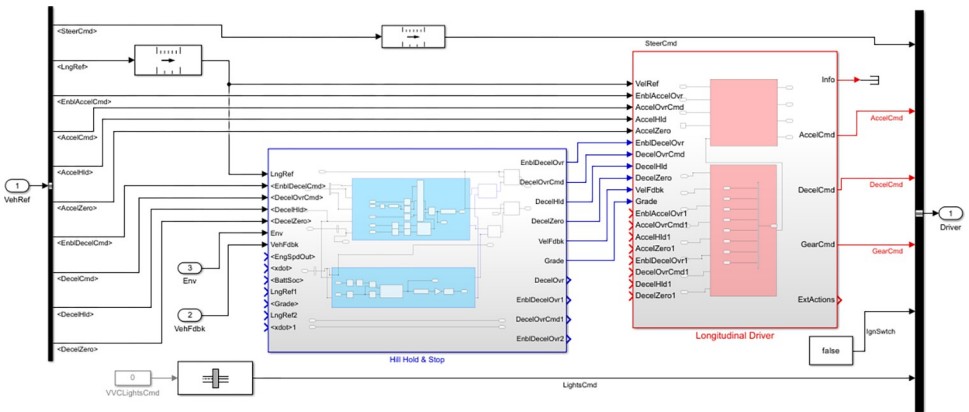

**Fig 7. The simulation of the Driver Commands block in longitudinal driver mode.**

The Controller block is a main component in the simulation, connected bilaterally with the Vehicle block. This block is comprised of seven subsystems named Vehicle Control Unit (VCU), Engine Control Unit (ECU), Transmission Control Unit (TCU), Brake Control Unit (BCU), Active Differential Control (ACD), Battery Management System (BMS), and Thermal Management (TM). VCU is a master-slave architecture designed to control the powertrain in vehicles. It is integrated into the engine controller ECU, which is capable of estimating the parameters of open-loop air, fuel, spark, and cam-phaser actuator from engine speed and torque signals. Gear shifting is predicted in TCU, which is determined from the inputs of deceleration, acceleration, engine speed, and instantaneous vehicle speed.

The Vehicle block is the most complex component in simulating the real-vehicle model, as it determines the performance of various parts of the vehicle, including the engine, drivetrain, body, suspension system, chassis, vehicle electronics, sensors, and pedal cluster. The engine subsystem is responsible for modeling the engine in terms of maximum torque, fuel flow, engine efficiency factor and engine speed. For the test vehicles in this study, the engine configuration was selected as Mapped Spark Ignition (SI), with the exception that vehicles #2 and #3 also feature turbochargers. The engine subsystem's basic inputs include engine speed, environment, vehicle body, and engine command. For each vehicle, a maximum torque curve is mapped to the model, which determines the maximum brake torque in terms of engine speed. The average brake specific fuel consumption and fuel specific gravity values are assumed to be equal to 350 g/kwh and 0.754, respectively. Other effective configurations in this subsystem include the number of cylinders, gas constant air, and air standard temperature, which are

**Table 3. Parameter values of Driver Commands block.**

| Parameters | Vehicle | | | | |
|---|---|---|---|---|---|
| | #1 | #2 | #3 | #4 | #5 |
| Mass | 1410 | 1414 | 1700 | 1315 | 1545 |
| Driver Response Time | 0.1 | 0.1 | 0.1 | 0.1 | 0.1 |
| Preview Distance | 4.1 | 4.1 | 4.1 | 4.1 | 4.1 |
| Rolling Resistance Coe. | 2.5 | 2.5 | 2.5 | 2.5 | 2.5 |
| Aerodynamic Drag Coe. | 0.25 | 0.30 | 0.45 | 0.30 | 0.30 |
| Gravitational Constant | 9.81 | 9.81 | 9.81 | 9.81 | 9.81 |
| Initial Gear | 0 | 0 | 0 | 0 | 0 |
| Gear Shift Delay | 0.11 | 0.12 | 0.10 | 0.12 | 0.12 |

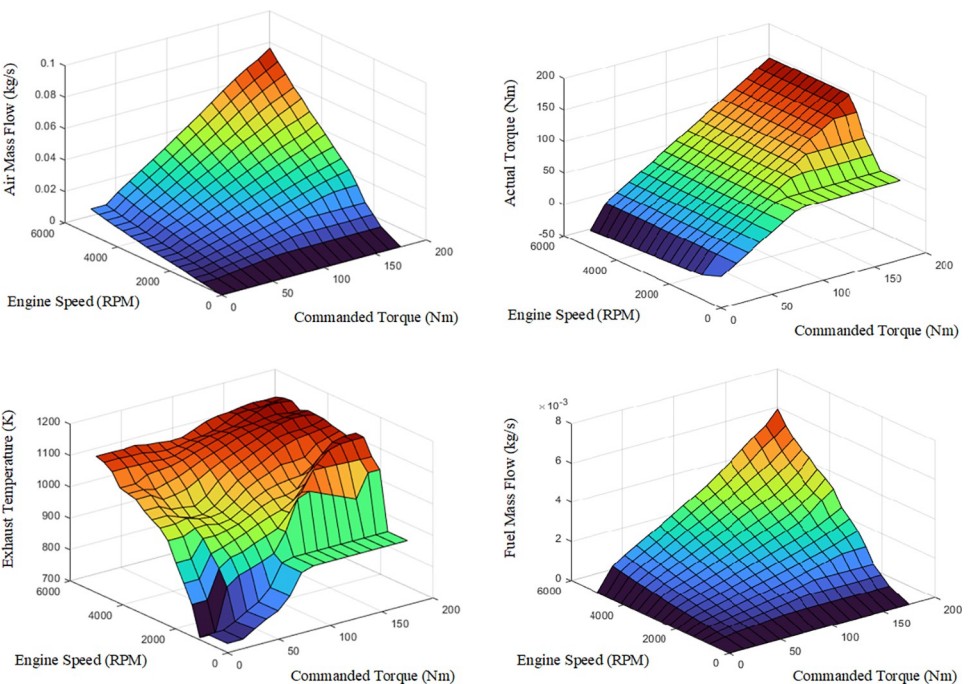

**Fig 8.** Displaying the parameters of (a) air mass flow map, (b) exhaust temperature map, (c) actual brake torque map, and (d) fuel mass flow map according to engine speed and commanded torque.

selected according to the characteristics of each vehicle. To simplify the analysis of results, the parameters of air mass flow, exhaust temperature, actual brake torque, and fuel mass flow in terms of engine speed and commanded torque are considered the same for all vehicles. The values of these parameters are shown in Fig 8.

The drivetrain subsystem consists of two parts: drivetrain layout and actuators, which are designed to model the powertrain and transmission of vehicles. The vehicle electronics subsystem is responsible for controlling the voltage and current of fuel cells. However, the presence of a HEV in this study has resulted in a significant difference between vehicle #5 and the other four conventional vehicles. This distinction is made up of the battery system and electric machines to supply the vehicle's energy according to Fig 9. The parameters specific to this vehicle are as follows: the number of cells in series is 96, the initial battery capacity is 3.18 Ampere-hours (Ah) and the rated capacity at nominal temperature is 5.3 Ah.

Finally, the output of the Vehicle block is fed back into the Scenarios and Environment blocks to ensure the stability of the system. After completing the model design, different drive cycles are applied in the Scenarios block, and then the model is run. The outputs of each port can be visualized using scopes in the Visualization block. It should be noted that the calculation of fuel consumption and emissions in this section is obtained through calibration maps in the Mapped SI Engine block. These maps model engine behavior and control parameters, which are defined as separate lookup table for each type of pollution (i.e., CO, HC, NOx, and $CO_2$). To be more precise, engine-out emissions and engine fuel flow obtained for each vehicle are a function of engine torque and speed, illustrating the engine's response to control inputs in different situations.

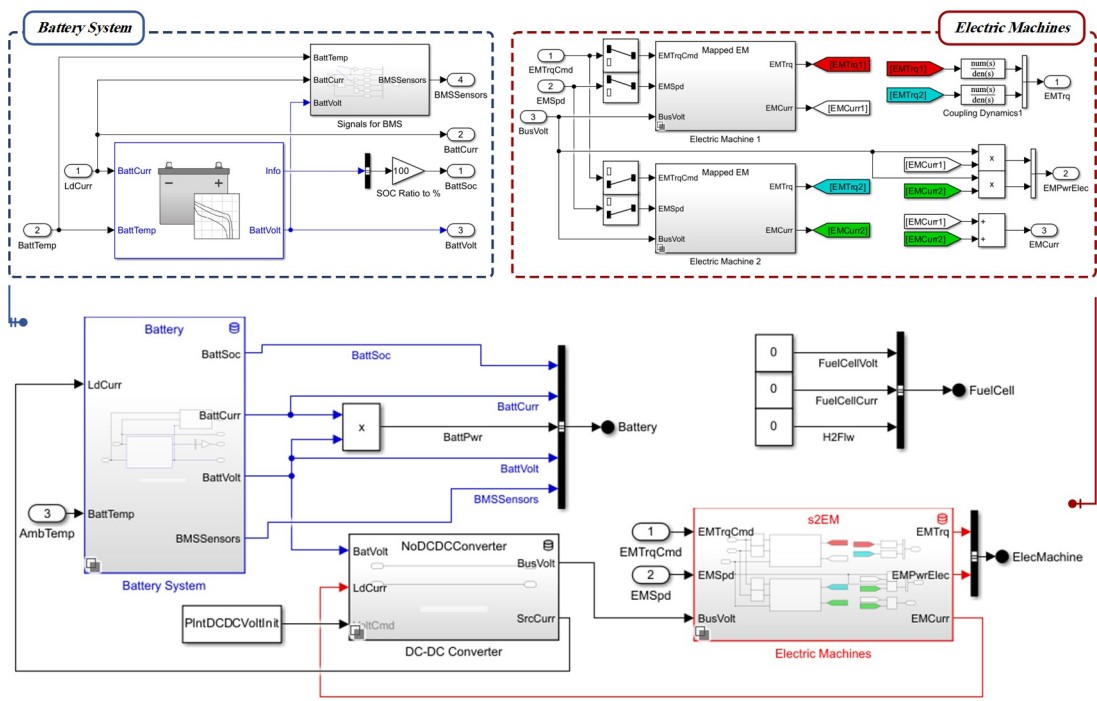

**Fig 9. Simulation of the vehicle electronics subsystem for vehicle #5.**

## Results and discussion

In this part, the driving cycles developed through K-means clustering and PCA algorithm are utilized to estimate the driving characteristic parameters, fuel consumption, and emissions of vehicles. Initially, the 12 driving characteristic parameters used in the PCA algorithm are calculated for the final driving cycles. Subsequently, by simulating the vehicles and inputting the driving cycles, fuel consumption and emissions are calculated. Finally, each vehicle is compared with other vehicles in terms of powertrain, engine downsizing, and body style based on their characteristics. To eliminate the influence of real-world driving conditions, the same procedure is repeated for the Federal Test Procedure 75 (FTP-75) and the Urban Dynamometer Driving Schedule (UDDS) cycles, using the same vehicle category [34, 35].

### Effect of driving characteristic parameters on fuel consumption and emissions

Driving behavior, traffic conditions, vehicle technology, and sample routes significantly influence a vehicle's driving characteristics, fuel consumption, and emissions. Each vehicle has its unique driving cycle, leading to different parameter values. Table 4 presents driving characteristic parameters, while Table 5 shows fuel consumption and emissions, measured in l/100km, g/km, and g/mi, respectively. The sample routes are identified by R1, R2, R3, and R4, which correspond to routes 1, 2, 3, and 4, respectively.

Fuel consumption and emissions are affected by multitude factors, with key contributors being traffic congestion, which leads to prolonged idling and reduced efficiency, and aggressive driving behaviors like excessive speed and acceleration. The vehicle's technology, including the engine management system and emission controls, also play a crucial role in determining fuel consumption and emissions. To illustrate the effects of these parameters on

**Table 4. Values of driving characteristic parameters.**

| Drive cycle | Drive time (%) | Idle time (%) | Cruise time (%) | Accelerate time (%) | Decelerate time (%) | Average driving speed (km/h) | Average trip speed (km/h) | Max speed (km/h) | Mean positive acceleration (m/s²) | Mean negative acceleration (m/s²) | Standard deviation speed | Standard deviation acceleration |
|---|---|---|---|---|---|---|---|---|---|---|---|---|
| #1-R1 | 90.92 | 9.07 | 9.24 | 38.42 | 43.25 | 36.97 | 33.62 | 77.29 | 0.69 | -0.62 | 10.77 | 0.91 |
| #1-R2 | 98.96 | 1.03 | 11.42 | 43.90 | 43.62 | 46.19 | 45.72 | 81.90 | 0.57 | -0.57 | 13.97 | 0.78 |
| #1-R3 | 82.23 | 17.76 | 7.33 | 35.69 | 39.19 | 25.81 | 21.20 | 75.67 | 0.73 | -0.68 | 8.39 | 0.89 |
| #1-R4 | 99.16 | 0.83 | 11.00 | 43.78 | 44.37 | 43.06 | 42.70 | 89.39 | 0.59 | -0.58 | 13.20 | 0.74 |
| #2-R1 | 90.49 | 9.50 | 10.34 | 37.86 | 42.28 | 33.55 | 30.38 | 70.31 | 0.73 | -0.65 | 9.65 | 0.95 |
| #2-R2 | 95.16 | 4.83 | 8.50 | 41.28 | 45.37 | 33.59 | 31.97 | 87.30 | 0.73 | -0.68 | 10.91 | 0.89 |
| #2-R3 | 91.49 | 8.50 | 9.34 | 40.28 | 41.87 | 40.10 | 36.68 | 85.90 | 0.75 | -0.73 | 12.18 | 0.99 |
| #2-R4 | 98.99 | 1.00 | 8.42 | 44.03 | 46.54 | 49.97 | 49.50 | 97.81 | 0.81 | -0.77 | 15.37 | 1.03 |
| #3-R1 | 87.23 | 12.76 | 11.09 | 37.03 | 39.11 | 35.03 | 30.56 | 76.90 | 0.53 | -0.53 | 10.59 | 0.81 |
| #3-R2 | 97.91 | 2.08 | 9.67 | 44.53 | 43.70 | 45.94 | 45.00 | 94.39 | 0.63 | -0.64 | 14.41 | 0.84 |
| #3-R3 | 88.49 | 11.50 | 10.59 | 38.11 | 39.78 | 35.60 | 31.50 | 77.98 | 0.67 | -0.65 | 10.64 | 0.97 |
| #3-R4 | 94.16 | 5.83 | 12.09 | 39.17 | 42.89 | 48.53 | 45.68 | 94.90 | 0.64 | -0.61 | 14.61 | 0.86 |
| #4-R1 | 84.40 | 15.59 | 8.84 | 38.86 | 36.69 | 43.92 | 37.08 | 98.57 | 0.56 | -0.58 | 12.99 | 0.77 |
| #4-R2 | 93.07 | 6.92 | 10.25 | 44.70 | 38.11 | 36.50 | 33.98 | 82.37 | 0.57 | -0.67 | 11.31 | 0.90 |
| #4-R3 | 87.57 | 12.42 | 8.75 | 37.78 | 41.03 | 21.71 | 19.01 | 60.59 | 0.69 | -0.66 | 6.61 | 0.96 |
| #4-R4 | 99.12 | 0.87 | 20.19 | 39.94 | 38.98 | 43.52 | 43.16 | 81.79 | 0.47 | -0.51 | 14.04 | 0.79 |
| #5-R1 | 90.40 | 9.59 | 17.43 | 37.19 | 35.77 | 39.82 | 36.00 | 99.90 | 0.74 | -0.77 | 12.59 | 0.88 |
| #5-R2 | 97.49 | 2.50 | 12.34 | 46.87 | 38.28 | 52.49 | 51.16 | 103.68 | 0.48 | -0.59 | 16.01 | 0.89 |
| #5-R3 | 88.74 | 11.25 | 12.17 | 39.61 | 36.94 | 35.78 | 31.75 | 72.68 | 0.54 | -0.60 | 10.56 | 0.86 |
| FTP-75 | 87.14 | 12.86 | 20.06 | 36.45 | 30.63 | 39.21 | 34.20 | 91.08 | 0.42 | -0.46 | 23.51 | 0.63 |
| UDDS | 86.19 | 13.81 | 18.04 | 36.96 | 31.19 | 36.60 | 31.60 | 91.15 | 0.43 | -0.46 | 21.46 | 0.64 |

**Table 5. The measure of fuel consumption and emissions.**

| Drive cycle | Fuel consumption (l/100km) | $CO_2$ (g/km) | HC (g/mi) | CO (g/mi) | $NO_X$ (g/mi) |
|---|---|---|---|---|---|
| #1-R1 | 10.04 | 230.00 | 0.069 | 0.171 | 0.109 |
| #1-R2 | 12.89 | 294.30 | 0.093 | 0.234 | 0.139 |
| #1-R3 | 15.68 | 356.40 | 0.112 | 0.370 | 0.174 |
| #1-R4 | 9.52 | 217.50 | 0.066 | 0.195 | 0.104 |
| #2-R1 | 11.33 | 227.83 | 0.703 | 26.557 | 0.002 |
| #2-R2 | 11.83 | 232.32 | 0.844 | 31.819 | 0.002 |
| #2-R3 | 11.13 | 215.69 | 0.880 | 32.934 | 0.002 |
| #2-R4 | 10.55 | 201.34 | 0.894 | 33.285 | 0.002 |
| #3-R1 | 12.20 | 242.04 | 0.838 | 33.053 | 0.001 |
| #3-R2 | 13.00 | 243.32 | 1.195 | 46.321 | 0.001 |
| #3-R3 | 13.20 | 267.64 | 1.226 | 47.750 | 0.001 |
| #3-R4 | 12.76 | 234.36 | 1.285 | 49.608 | 0.001 |
| #4-R1 | 7.83 | 179.20 | 0.048 | 0.087 | 0.082 |
| #4-R2 | 9.66 | 221.70 | 0.063 | 0.106 | 0.099 |
| #4-R3 | 12.17 | 278.50 | 0.078 | 0.135 | 0.121 |
| #4-R4 | 7.91 | 182.00 | 0.050 | 0.081 | 0.079 |
| #5-R1 | 7.85 | 157.50 | 0.215 | 11.434 | 0.001 |
| #5-R2 | 7.35 | 148.20 | 0.165 | 8.940 | 0.001 |
| #5-R3 | 7.30 | 160.20 | 0.146 | 8.133 | 0.001 |

fuel consumption and emissions, the driving behavior of five vehicles on four sample routes were examined.

The results indicate that vehicles with lower traffic congestion (e.g., shorter idling times) exhibit lower fuel consumption and emissions. For instance, vehicle #1 on sample route 4 had only 0.83% idling time, resulting in lower fuel use and emissions. Extended idling can consume more fuel than restarting; however, restarting increases CO, HC, and NOx emissions [36]. Interestingly, vehicle #4 on route 1 had the highest idling percentage (15.59%) but lower emissions, likely due to low acceleration and an average speed of 43.92 km/h.

Turbocharged vehicle #2 on sample route 4, with only 1% idling time and a high acceleration rate (44.03%), showed increased CO and HC emissions, despite relatively low fuel consumption and $CO_2$ emissions, possibly due to turbocharging at high speeds. For this turbocharged vehicle, an average driving speed tend to exhibit higher emissions of CO and HC, whereas those with lower average driving speeds tend to exhibit lower emissions [21]. Vehicle #3, a turbocharged SUV, displayed this trend as well. On sample route 1, it faced heavy traffic, resulting in lower average speed (35.03 km/h) and higher fuel consumption and $CO_2$ emissions [37].

Other driving characteristic parameters, such as the standard deviation of speed and acceleration, and idling time, correlate significantly with fuel consumption and emissions. Increased idling time and higher standard deviations of acceleration and speed are associated with increased fuel consumption and emissions in conventional vehicles. Conversely, a higher standard deviation of speed corresponds to lower CO and HC emissions in conventional vehicles, while the trend is reversed for turbocharged vehicles.

Vehicle #5 uniquely reduces fuel consumption during idling [38]. On sample route 3, it had longer idle times and lower average speeds, leading to decreased fuel consumption, CO, and HC emissions, though $CO_2$ emissions were higher.

As NOx pollution is relatively low and comparable across vehicles, it has not been individually assessed for each vehicle. The results indicate that aggressive driving behaviors contribute to increased NOx emissions [39]. Furthermore, frequent instances of stopping the engine while idling have been observed to result in higher NOx emissions.

## Effect of powertrain on fuel consumption and emissions

To assess the impact of powertrain type on fuel consumption and emissions, a comparative study was conducted on three vehicles: a gasoline vehicle (vehicle #1), a hybrid vehicle (vehicle #5), and a dual fuel vehicle (vehicle #4). Results are illustrated in the Box & Whisker diagram in Fig 10 and summarized in Table 6.

The hybrid vehicle shows significant advantages, with approximately 39% lower fuel consumption than gasoline and 26% lower than dual fuel vehicles due to efficient energy capture during deceleration [40]. It also exhibits 34% less $CO_2$ emissions than vehicle #1 and 10% less than vehicle #4, according to engine shutdown during idling and the use of electric energy sources [40]. Unique subsystems like brushless motors and planetary gearboxes enhance energy efficiency.

Vehicle #4, the dual fuel option, demonstrates an 18% improvement in fuel efficiency compared to gasoline vehicles. Compressed Natural Gas (CNG) has a lower carbon content, resulting in about 28% lower $CO_2$ emissions compared to vehicle #1 [41, 42].

In terms of emissions, the hybrid vehicle emits more CO than gasoline and dual fuel vehicles due to increased power demands on the internal combustion engine. Primarily due to the increased power demands placed on internal combustion engines in hybrid electric vehicles, which are necessitated by the simultaneous powering of both wheels and batteries. This may

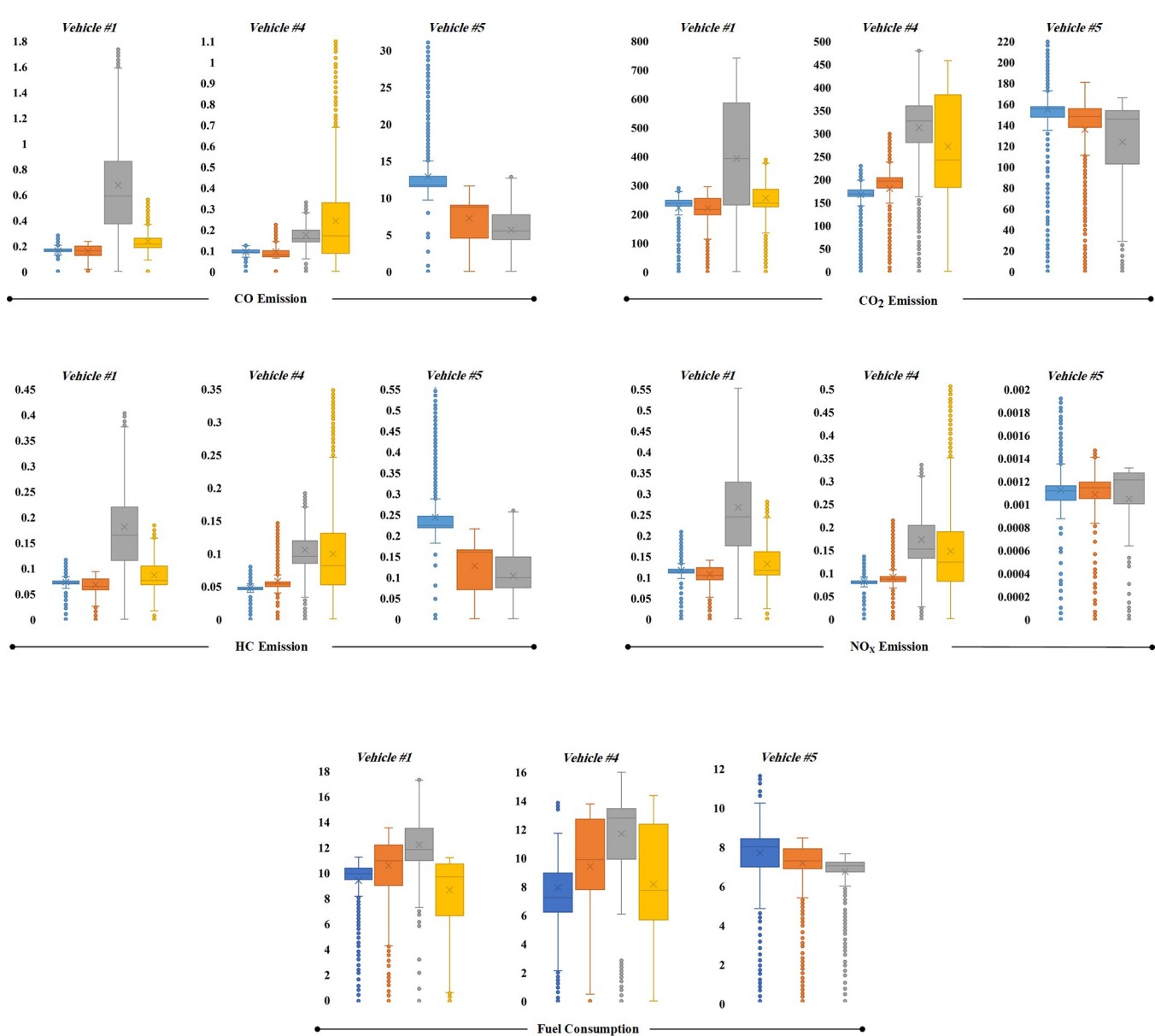

**Fig 10. Comparison of powertrain effect on fuel consumption and emission over sample routes.**

**Table 6. Measures of fuel consumption and pollutant emissions for standard driving cycles of powertrain comparison.**

| Drive cycle | Fuel consumption (l/100km) | $CO_2$ (g/km) | HC (g/mi) | CO (g/mi) | $NO_X$ (g/mi) |
|---|---|---|---|---|---|
| #1-FTP-75 | 7.96 | 183.6 | 0.054 | 0.090 | 0.071 |
| #1-UDDS | 8.16 | 200.5 | 0.069 | 0.094 | 0.058 |
| #4- FTP-75 | 7.38 | 127.4 | 0.029 | 0.089 | 0.001 |
| #4- UDDS | 7.87 | 97.5 | 0.001 | 0.090 | 0.001 |
| #5- FTP-75 | 5.64 | 169.4 | 0.049 | 0.085 | 0.067 |
| #5- UDDS | 4.52 | 181.0 | 0.058 | 0.056 | 0.054 |

also be attributed to the shorter running time and longer intervals between engine cycles, which may result in inadequate engine warm-up, thereby contributing to increased CO emissions [18]. In contrast, HC emissions are lower in hybrids under standard cycles but can increase under real conditions. Gasoline vehicles emit 45% more CO and 39% more HC than dual fuel vehicles, owing to the more complete combustion with CNG.

The findings also demonstrate that the hybrid vehicle exhibits a notable reduction in NOx emissions compared to conventional vehicles. This reduction is attributed to the use of the electric motor in the hybrid engine, which leads to lower power output and subsequently reduced in cylinder combustion temperature. Elevated temperature is a critical factor in NOx generation, and reducing the temperature leads to a decrease in NOx production.

## Effect of engine downsizing on fuel consumption and emissions

The compression ratio and turbocharging significantly influence the operation and emissions of internal combustion engines [43]. Engine downsizing, which replaces larger naturally aspirated engines with smaller turbocharged ones, improves fuel efficiency and reduces emissions. Turbocharging enhances volumetric efficiency by using exhaust waste heat for air compression [44]. Fig 11 compares naturally aspirated and turbocharged engines, while Table 7 presents similar results for standard driving cycles. Turbocharger activation depends on driving conditions, resulting in comparable fuel consumption and $CO_2$ emissions between vehicles.

This study shows that engine downsizing in vehicle #2 leads to better fuel efficiency, except for higher consumption in sample routes 1 and 4. This aligns with previous research indicating that downsizing can cut fuel use and emissions [45], although it depends on traffic and speed. The results also confirm that reducing the capacity of turbocharged engines lowers $CO_2$ emissions [44, 46]. In all driving cycles, $CO_2$ emissions were higher in naturally aspirated engines,

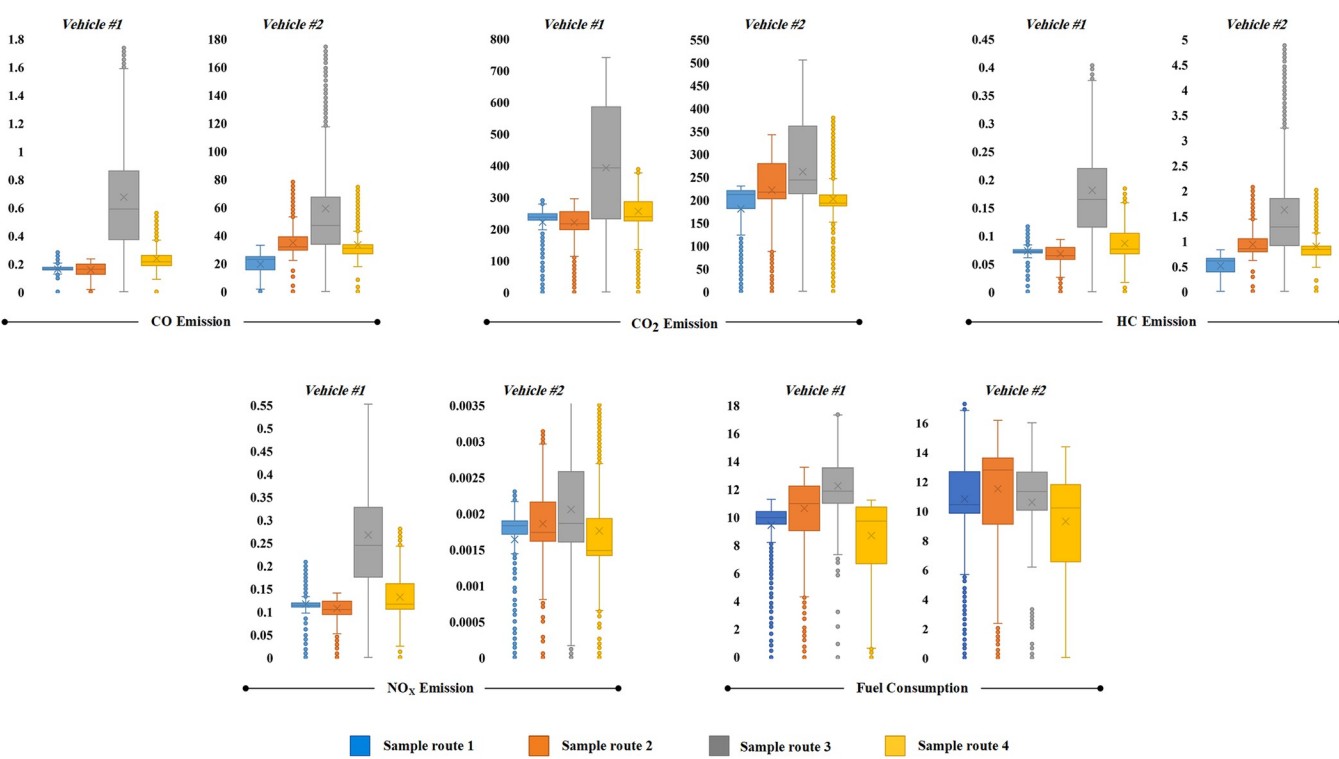

**Fig 11. The comparison of engine downsizing effect on fuel consumption and emission over sample routes.**

Table 7. Measures of fuel consumption and pollutant emissions for standard driving cycles of engine downsizing comparison.

| Drive cycle | Fuel consumption (l/100km) | $CO_2$ (g/km) | HC (g/mi) | CO (g/mi) | $NO_X$ (g/mi) |
|---|---|---|---|---|---|
| #1-FTP-75 | 7.96 | 183.60 | 0.054 | 0.090 | 0.071 |
| #1-UDDS | 8.16 | 200.50 | 0.069 | 0.094 | 0.058 |
| #2- FTP-75 | 7.73 | 159.28 | 0.389 | 14.365 | 0.001 |
| #2- UDDS | 7.98 | 166.58 | 0.392 | 14.423 | 0.001 |

while NOx emissions were over 90% greater in naturally aspirated engines compared to turbocharged vehicles [46]. The benefits of turbocharging in reducing NOx emissions are well-established, while simultaneously enhancing power density [47]. This is due to the fact that turbocharging allows for more efficient combustion, thereby reducing the formation of NOx emissions.

For CO and HC emissions, turbocharged engines typically emit more CO and HC than naturally aspirated ones [48]. This is due to their design for high-speed performance. In urban and rural settings, naturally aspirated vehicles produce lower CO emissions because of their operation at lower speeds and loads. Conversely, turbocharged engines generate more CO during motorway driving due to higher loads and speeds. These findings emphasize the need to consider specific driving conditions and engine characteristics when assessing the effects of engine downsizing on fuel consumption and emissions.

## Effect of body style on fuel consumption and emissions

To compare the effects of body style on fuel consumption and emissions, we selected vehicle #3, an SUV, and vehicle #2, a sedan, both equipped with turbocharged engines. Notably, vehicle #3 features an automatic transmission, while vehicle #2 is equipped with a manual transmission. The comparative analysis of these vehicles under real driving cycles is illustrated in Fig 12, and the summary of their fuel consumption and emissions under standardized driving cycles is presented in Table 8.

It is evident that the sedan vehicle exhibited lower fuel consumption, HC, CO, and $CO_2$ emissions compared to the SUV, with a reduction of 13% for fuel consumption, 14% for HC, 20% for CO, and 15% for $CO_2$. The disparity in fuel consumption and emissions between the two vehicles can be attributed to several factors [49]. One of the primary factors contributing to this disparity is the weight difference. SUVs typically have a greater weight than sedans, which results in an increased load on the engine. This increased load is compensated by stronger engine power, leading to higher fuel consumption. Furthermore, SUVs tend to have larger engine volumes and higher volumes of exhaust gases, which also contribute to increased fuel consumption and emissions.

Additionally, the design characteristics of SUVs, including a taller cabin and larger wheel radius, hinder aerodynamic optimization. The increased wind resistance and higher drag coefficient associated with SUVs exacerbate fuel consumption and emissions.

Interestingly, under standardized driving cycles, vehicle #2 exhibited elevated levels of HC and CO emissions as compared to vehicle #3. This can largely be attributed to the cold start phase of the driving cycles. During the FTP-75 and UDDS cycles, high speeds are encountered shortly after starting, which can affect driver behavior and lead to gear shift delays. Furthermore, the engine's operation during cold starts often necessitates the use of an enriched fuel mixture to prevent incomplete combustion, subsequently resulting in increased HC and CO emissions during this phase.

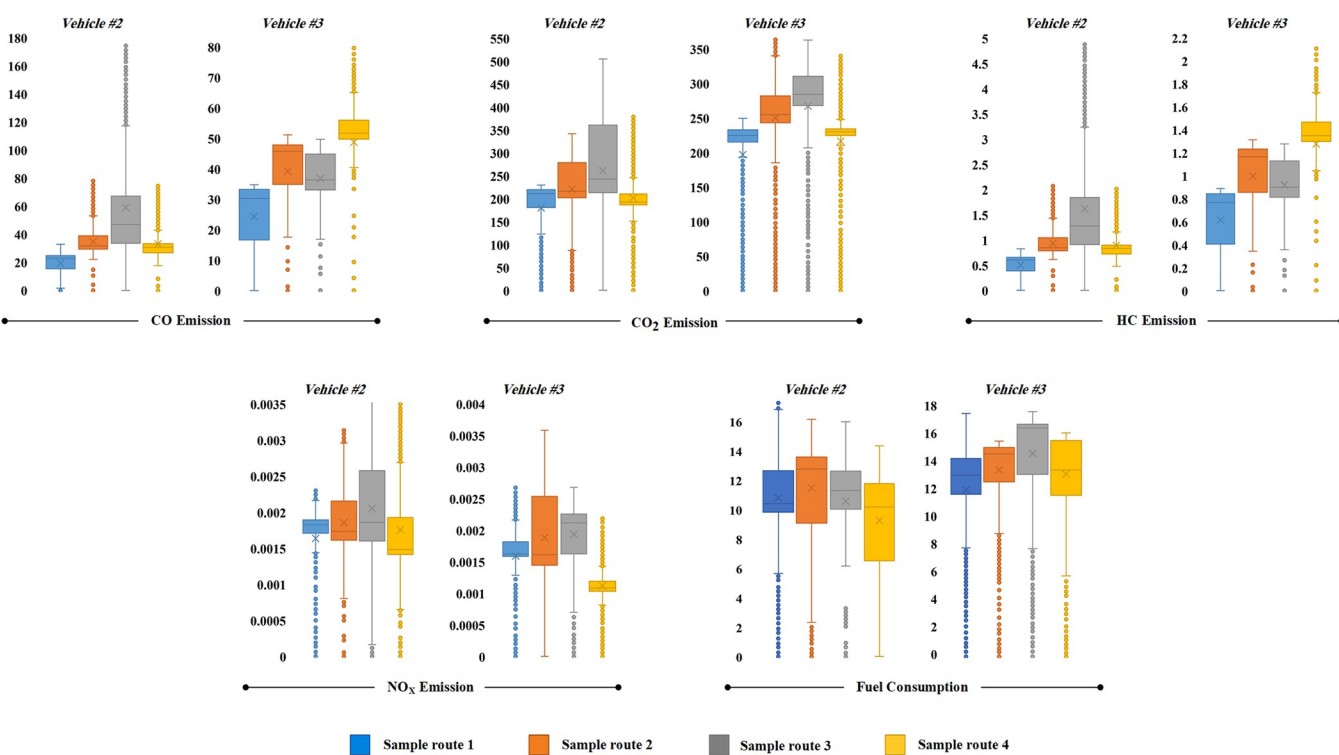

**Fig 12. Comparison of body style effect on fuel consumption and emission over sample routes.**

## Conclusions and future works

This study aimed to investigate the driving characteristic parameters, fuel consumption, and emissions performance of vehicles equipped with various technologies, including powertrain, engine, and body style, under both real and standard driving cycles. A total of 19 drive cycles were created using the K-means and PCA algorithms, based on the selection of four sample routes. The vehicles were modeled in MATLAB/Simulink based on their specifications, and real and standard driving cycles were simulated to measure their fuel consumption and emissions. The major findings from comparing the driving characteristic parameters, fuel consumption, and emissions of four conventional vehicles and one hybrid vehicle are as follows:

- It was found that driving characteristic parameters exhibit distinct relationships with fuel consumption and emissions under various conditions and vehicle technologies.

- Gasoline and dual fuel vehicles exhibit similar behaviors under traffic conditions, with the exception of slightly higher fuel consumption (approximately 18%) and increased emissions in the gasoline vehicle. In contrast, hybrid vehicle demonstrated an advantage in fuel consumption during idle time, with reductions of around 39% and 26% compared to gasoline and dual fuel vehicles, respectively.

**Table 8. Measures of fuel consumption and pollutant emissions for standard drive cycles of body style comparison.**

| Drive cycle | Fuel consumption (l/100km) | $CO_2$ (g/km) | HC (g/mi) | CO (g/mi) | $NO_X$ (g/mi) |
|---|---|---|---|---|---|
| #2-FTP-75 | 7.73 | 159.28 | 0.389 | 14.365 | 0.001 |
| #2-UDDS | 7.98 | 166.58 | 0.392 | 14.423 | 0.001 |
| #3- FTP-75 | 8.84 | 199.68 | 0.114 | 4.471 | 0.002 |
| #3- UDDS | 8.13 | 188.93 | 0.001 | 1.575 | 0.001 |

- The turbocharged vehicles yield a notable benefit in terms of fuel consumption and $CO_2$ emissions, particularly at high speeds where the turbocharger is activated. In this scenario, the turbocharged vehicle consumes approximately 6% less fuel and 19% less $CO_2$ than its naturally aspirated vehicle.

- Regarding specific vehicle technologies, it was observed that hybrid powertrain and the presence of turbocharger had the highest emissions in terms of CO and HC, while exhibiting lower NOx emissions.

- The results indicate that SUV with turbocharging technology exhibit lower fuel consumption at high cruising speeds, although the magnitude of this advantage is relatively higher, about 13% compared to sedans. The turbocharged vehicles exhibit elevated levels of CO and HC emissions in real-world driving cycles compared to standard driving cycles, which are characterized by more aggressive driving patterns.

Future research recommendations include capturing real-world driving behavior by incorporating driving data from multiple individuals, potentially through the use of a chasing method, especially in high traffic areas such as metropolitan. Additionally, exploring emerging technologies such as fully electric powertrains, evaluating different types of transmissions, and assessing the impact of Exhaust Gas Recirculation (EGR). Ultimately, these findings not only underscore the need for refined policies and regulations based on real-world data but also highlight the potential for innovative vehicle technologies and infrastructure improvements to facilitate the transition to greener alternatives in urban transportation.

## Author Contributions

**Conceptualization:** Masoud Masih Tehrani, Mohammad Azadi, Ashkan Moosavian.

**Data curation:** Elmira Bagheri, Mohammad Azadi.

**Formal analysis:** Elmira Bagheri, Masoud Masih Tehrani.

**Funding acquisition:** Mohammad Azadi.

**Investigation:** Elmira Bagheri, Masoud Masih Tehrani, Mohammad Azadi.

**Methodology:** Elmira Bagheri, Masoud Masih Tehrani, Mohammad Azadi.

**Project administration:** Masoud Masih Tehrani, Mohammad Azadi.

**Resources:** Elmira Bagheri, Masoud Masih Tehrani, Mohammad Azadi, Ashkan Moosavian.

**Software:** Elmira Bagheri.

**Supervision:** Masoud Masih Tehrani, Mohammad Azadi, Ashkan Moosavian.

**Validation:** Masoud Masih Tehrani, Mohammad Azadi.

**Visualization:** Elmira Bagheri, Masoud Masih Tehrani, Mohammad Azadi.

**Writing – original draft:** Elmira Bagheri.

**Writing – review & editing:** Masoud Masih Tehrani, Mohammad Azadi, Ashkan Moosavian.

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
