## [Decision Letter · Decision Letter 0]

1 May 2024

PONE-D-24-06667Impact of driving characteristic parameters and vehicle type on fuel consumption and emissions performance over real driving cyclesPLOS ONE

Dear Dr. Azadi,

Thank you for submitting your manuscript to PLOS ONE. After careful consideration, we feel that it has merit but does not fully meet PLOS ONE’s publication criteria as it currently stands. Therefore, we invite you to submit a revised version of the manuscript that addresses the points raised during the review process.

We look forward to receiving your revised manuscript.

Kind regards,

Jose Balsa-Barreiro

Academic Editor

PLOS ONE

Journal Requirements:

2. You indicated that ethical approval was not necessary for your study, although the study appears to report data collected about a human participant while driving. We understand that the framework for ethical oversight requirements for studies of this type may differ depending on the setting and we would appreciate some further clarification regarding your research. Could you please provide further details on why your study is exempt from the need for approval and confirmation from your institutional review board or research ethics committee (e.g., in the form of a letter or email correspondence) that ethics review was not necessary for this study? Please include a copy of the correspondence as an ""Other"" file.

4. We note that Figure 2 in your submission contain map images which may be copyrighted. All PLOS content is published under the Creative Commons Attribution License (CC BY 4.0), which means that the manuscript, images, and Supporting Information files will be freely available online, and any third party is permitted to access, download, copy, distribute, and use these materials in any way, even commercially, with proper attribution. For these reasons, we cannot publish previously copyrighted maps or satellite images created using proprietary data, such as Google software (Google Maps, Street View, and Earth). For more information, see our copyright guidelines: http://journals.plos.org/plosone/s/licenses-and-copyright.

We require you to either present written permission from the copyright holder to publish these figures specifically under the CC BY 4.0 license, or remove the figures from your submission:

Additional Editor Comments:

Dear Author,

After the reviews conducted by two independent experts on the proposed topic, my recommendation is to address all the concerns that remain outstanding according to one of the reviewers (major revisions/rejection). Once these concerns have been addressed satisfactorily, a new round of reviews will be conducted, involving a third reviewer.

One aspect I have observed is that the authors do not include a research methodology that is highly relevant for this type of study on current driving performance: naturalistic driving experiments. Among other studies, the authors may consider evaluating research related to the Extraction of naturalistic driving patterns or the Representation of kinematic driving behavior using Geographic Information Systems, where driving patterns of individuals under real conditions are precisely addressed and visualized.

My best regards,

The Assoc. Editor

Reviewers' comments:

Reviewer's Responses to Questions

**Comments to the Author**

1. Is the manuscript technically sound, and do the data support the conclusions?

Reviewer #1: Yes

Reviewer #2: Partly

2. Has the statistical analysis been performed appropriately and rigorously? 

Reviewer #1: Yes

Reviewer #2: No

3. Have the authors made all data underlying the findings in their manuscript fully available?

Reviewer #1: Yes

Reviewer #2: No

4. Is the manuscript presented in an intelligible fashion and written in standard English?

Reviewer #1: Yes

Reviewer #2: Yes

5. Review Comments to the Author

**Reviewer #1:** The manuscript proposes "Impact of driving characteristic parameters and vehicle type on fuel consumption and

emissions performance over real driving cycles."

The reviewer's concerns are as follows:

1- The novelty of this study is not clear. To enable readers to follow the article more quickly and effectively, a flowchart summarizing the article should be added to the introduction section.

2- Authors have not provided comparison between their proposed methodology and the previous ones in the introduction. According to the current introduction, there is no contribution in the current study.

3- It is recommended to add the following reference.

a)Adak, S., Cangi, H., Kaya, R., Yılmaz, A. S. (2022). Effects of Electric Vehicles and Charging Stations on Microgrid Power Quality. Gazi University Journal of Science Part A: Engineering and Innovation, 9(3), 276-286. https://doi.org/10.54287/gujsa.1153313

**Reviewer #2:** This paper intended to consider the driving characteristic parameters, fuel consumption, and

emissions performance of vehicles with different technologies such as powertrain, engine, and body style under both real and standard drive cycles. 19 drive cycles were created by utilizing the K-means and PCA algorithms, based on the selection of four sample routes. The vehicles underwent modeling in MATLAB/Simulink based on their specifications, and real and standard drive cycles were simulated to measure their fuel consumption and emissions.

Vehicles #1, 2, 3, and 4 are conventional vehicles and vehicle #5 is a hybrid vehicle considered. The way to choose vehicles considered different types of powertrains, body styles, and the presence of turbocharged. For driving characteristic parameters, emissions, and fuel consumption comparisons, three vehicle models for three powertrains (gasoline, hybrid electric, and dual fuel), two types of engines (turbocharged and naturally aspirated), and two vehicle body styles (i.e., sedan, SUV), are used.

To reduce the influence of the driver's behavior on the test results, the same driver drove all routes.

Even the driving cycles for 5 vehicles on the same route were very different from each other. (Fig.3 and Fig.5). This will negatively affect the analysis to be performed. No explanation was given for this.

Vehicle simulation steps performed in the MATLAB/Simulink program are not detailed enough. How are fuel consumption and emissions calculated? What is the difference between the calculations for a hybrid vehicle (Toyota Prius) and a dual fuel vehicle and the calculations for a conventional vehicle?

The result, which includes the effect of driving characteristic parameters, powertrain type, engine downsizing and body style on fuel consumption and emissions, was not found scientifically sufficient.

6. PLOS authors have the option to publish the peer review history of their article (what does this mean?). If published, this will include your full peer review and any attached files.

Reviewer #1: No

Reviewer #2: No

---

## [Author Response · Author response to Decision Letter 0]

30 Jun 2024

Please check the separated file for the answers to comments.

---

## [Decision Letter · Decision Letter 1]

20 Sep 2024

PONE-D-24-06667R1Impact of driving characteristic parameters and vehicle type on fuel consumption and emissions performance over real driving cyclesPLOS ONE

Dear Dr. Azadi,

Thank you for submitting your manuscript to PLOS ONE. After careful consideration, we feel that it has merit but does not fully meet PLOS ONE’s publication criteria as it currently stands. Therefore, we invite you to submit a revised version of the manuscript that addresses the points raised during the review process.

One key critique refers to what one of the reviewers mentioned about "How to integrate real and simulated data." Additionally, the authors introduce the naturalistic research method (whose methodology and objectives align with and complement what is presented here), but do not include any relevant studies in the reference list. Some suggestions were provided in the previous round. Furthermore, there is a lack of adequate cartographic treatment in some of the maps presented, where scale information is missing, among other deficiencies.

We look forward to receiving your revised manuscript.

Kind regards,

Jose Balsa-Barreiro

Academic Editor

PLOS ONE

Journal Requirements:

Additional Editor Comments:

Dear Authors,

After the extensive review process carried out by a broad group of reviewers, my recommendation is for an additional revision. While a significant number of experts suggest acceptance, two of them propose an additional revision, recommending major corrections, with one even suggesting rejection. Therefore, my proposal is that the authors revise the manuscript again, addressing the comments and suggestions made by the two dissenting reviewers. One key critique refers to what one of the reviewers mentioned about "How to integrate real and simulated data." Additionally, the authors introduce the naturalistic research method (whose methodology and objectives align with and complement what is presented here), but do not include any relevant studies in the reference list. Furthermore, there is a lack of adequate cartographic treatment in some of the maps presented, where scale information is missing, among other deficiencies.

With my best regards,

The Associate Editor

Reviewers' comments:

Reviewer's Responses to Questions

**Comments to the Author**

1. If the authors have adequately addressed your comments raised in a previous round of review and you feel that this manuscript is now acceptable for publication, you may indicate that here to bypass the “Comments to the Author” section, enter your conflict of interest statement in the “Confidential to Editor” section, and submit your "Accept" recommendation.

Reviewer #3: All comments have been addressed

Reviewer #4: (No Response)

Reviewer #5: All comments have been addressed

Reviewer #6: All comments have been addressed

Reviewer #7: All comments have been addressed

Reviewer #8: All comments have been addressed

Reviewer #9: (No Response)

2. Is the manuscript technically sound, and do the data support the conclusions?

Reviewer #3: Yes

Reviewer #4: Yes

Reviewer #5: Yes

Reviewer #6: Yes

Reviewer #7: Yes

Reviewer #8: Partly

Reviewer #9: Yes

3. Has the statistical analysis been performed appropriately and rigorously? 

Reviewer #3: Yes

Reviewer #4: Yes

Reviewer #5: Yes

Reviewer #6: Yes

Reviewer #7: Yes

Reviewer #8: Yes

Reviewer #9: Yes

4. Have the authors made all data underlying the findings in their manuscript fully available?

Reviewer #3: Yes

Reviewer #4: Yes

Reviewer #5: Yes

Reviewer #6: Yes

Reviewer #7: Yes

Reviewer #8: Yes

Reviewer #9: Yes

5. Is the manuscript presented in an intelligible fashion and written in standard English?

Reviewer #3: Yes

Reviewer #4: Yes

Reviewer #5: Yes

Reviewer #6: Yes

Reviewer #7: Yes

Reviewer #8: Yes

Reviewer #9: Yes

6. Review Comments to the Author

Reviewer #3: Comments have been addressed. Thank you. I noticed in the GitHub repository, one of the fields is in ".rar" format. Is that containing code or data? if so, please specify and consider uploading the files in a cloud environment (google drive, microsoft onedrive, or dropbox) and provide the link here in the github repository. This way, everyone will be able to access the files.

Reviewer #4: The reviewer's concerns are as follows:

1) The data in the manuscript was initiated by TTO several years ago, is it with timeliness? And only one driver was recruited to conduct the experiment, did the driver know the proficiency level of these different types of vehicles?

2) Figure 3 shows that there is little difference in the percentage of variance of driving characteristics parameters. Please mark the specific percentage values at the top to make it clear.

3) Please explain in details why choose these 12 driving characteristics and why reduce them to two sensitive parameters, why not 3 or 4···?

4) In the result, the comparisons among driving characteristics parameters, powertrain type, engine miniaturization and body style on fuel consumption and emissions are too repetitive, should be optimized more briefly.

5) A flow chart should be added to show the details on the vehicle simulation steps executed in the MATLAB/Simulink program.

Reviewer #5: (No Response)

Reviewer #6: (No Response)

Reviewer #7: I think all the answers to our reviewers have been well done.

This paper presents excellent ways to reduce exhaust emissions overall in order to implement eco-friendly policies for vehicles.

Reviewer #8: The manuscript proposes "Impact of driving characteristic parameters and vehicle type on fuel consumption and emissions performance over real driving cycles." The article shows a lack of innovation, detailed comments are as follows,

1. Is it too little data based on 4 traditional vehicles and 1 hybrid vehicle, and can it reflect driving characteristics? Are the drivers of the 5 cars the same? Is the driver consistent for each driving cycle? The driving style is only reflected in speed and acceleration, which is very one-sided. In addition, vehicle color, as a parameter, is also a variable, but it can only have 5 values. The same problem applies to other parameters, and limited data cannot lead to convincing conclusions.

2. The theoretical innovation of the article is not obvious, and it mentions real driving cycles, which were simulated using MATLAB. How to integrate real and simulated data? Moreover, in the simulation, there is no detailed introduction on how to distinguish between traditional vehicles and hybrid vehicles, and how to set the simulation environment and parameters.

3. If all 12 driving characteristic parameters are related to speed and acceleration, then in each driving cycle, if the driver is the same, this style will be severely correlated with the driver's driving style, which will lead to significant unreliability in the result analysis.

Reviewer #9: 1. Data Collection Frequency: The data collection frequency is set at 1 Hz. How does this frequency impact the accuracy of the driving characteristic parameters, especially during high-speed maneuvers?

2. Clustering Technique Justification: The authors employed the K-means algorithm for clustering micro-trips. What specific characteristics of the data justified the choice of K-means over other clustering methods?

3. Parameter Selection: The study identifies 12 driving characteristic parameters. What criteria were used to select these parameters, and how do they correlate with fuel consumption and emissions?

4. Impact of Weather Conditions: While data was collected under sunny conditions, how might variations in weather (e.g., rain, snow) affect the driving cycles and the resulting fuel consumption and emissions?

5. Sample Size and Representativeness: The study examines five vehicles. Is this sample size sufficient to draw generalizable conclusions about vehicle performance across different models and brands?

6. Future Research Directions: The authors mention the need for further research. What specific areas do they suggest for future studies to build upon their findings?

7. Practical Implications: What practical implications do the authors foresee from their findings for policymakers and automotive manufacturers in terms of improving vehicle technologies and urban planning?

7. PLOS authors have the option to publish the peer review history of their article (what does this mean?). If published, this will include your full peer review and any attached files.

Reviewer #3: **Yes: **Morteza Maleki

Reviewer #4: No

Reviewer #5: No

Reviewer #6: **Yes: **Dr. Zainab Ahmed Alkaissi

Reviewer #7: No

Reviewer #8: No

Reviewer #9: No

---

## [Author Response · Author response to Decision Letter 1]

8 Nov 2024

Please check the separated file provided for answers to comments.

---

## [Editor Report · Decision Letter 2]

25 Nov 2024

PONE-D-24-06667R2Impact of driving characteristic parameters and vehicle type on fuel consumption and emissions performance over real driving cyclesPLOS ONE

Dear Dr. Azadi,

Thank you for submitting your manuscript to PLOS ONE. After careful consideration, we feel that it has merit but does not fully meet PLOS ONE’s publication criteria as it currently stands. Therefore, we invite you to submit a revised version of the manuscript that addresses the points raised during the review process.

The inclusion of naturalistic driving in this study is a well-justified and appropriate approach, as it aligns closely with the need to capture real-world driving behaviors that significantly influence fuel consumption and emission levels. Naturalistic driving allows for the extraction of authentic driving patterns, which can then be correlated with specific vehicle performance metrics, providing a comprehensive understanding of the variability introduced by actual road conditions and driving styles. To strengthen the impact of this work, it is recommended to incorporate references to relevant studies that have successfully utilized methodologies such as Geographic Information Systems (GIS) or Global Navigation Satellite Systems (GNSS) to extract and analyze driving patterns. These studies demonstrate the critical role of spatial and kinematic data in representing driving behavior and its subsequent relationship to emissions and fuel efficiency (Extraction of naturalistic driving patterns or Representation of kinematic driving behaviour). Highlighting these methodologies can further underscore the robustness of the current approach while situating the work within the broader scientific context.

We look forward to receiving your revised manuscript.

Kind regards,

Jose Balsa-Barreiro

Academic Editor

PLOS ONE

Journal Requirements:

Additional Editor Comments:

Dear Authors,

The inclusion of naturalistic driving in this study is a well-justified and appropriate approach, as it aligns closely with the need to capture real-world driving behaviors that significantly influence fuel consumption and emission levels. Naturalistic driving allows for the extraction of authentic driving patterns, which can then be correlated with specific vehicle performance metrics, providing a comprehensive understanding of the variability introduced by actual road conditions and driving styles. To strengthen the impact of this work, it is recommended to incorporate references to relevant studies that have successfully utilized methodologies such as Geographic Information Systems (GIS) or Global Navigation Satellite Systems (GNSS) to extract and analyze driving patterns. These studies demonstrate the critical role of spatial and kinematic data in representing driving behavior and its subsequent relationship to emissions and fuel efficiency (Extraction of naturalistic driving patterns or Representation of kinematic driving behaviour). Highlighting these methodologies can further underscore the robustness of the current approach while situating the work within the broader scientific context.

With my best regards, the Assoc. Editor

---

## [Author Response · Author response to Decision Letter 2]

13 Dec 2024

The separated file was provided for the answers to the comments.

---

## [Editor Report · Decision Letter 3]

22 Dec 2024

Impact of driving characteristic parameters and vehicle type on fuel consumption and emissions performance over real driving cycles

PONE-D-24-06667R3

Dear Dr. Azadi,

We’re pleased to inform you that your manuscript has been judged scientifically suitable for publication and will be formally accepted for publication once it meets all outstanding technical requirements.

Kind regards,

Jose Balsa-Barreiro

Academic Editor

PLOS ONE

Additional Editor Comments (optional):

Dear Authors,

I am pleased to inform you that, after careful review of the revisions you have submitted, the comments and suggestions provided by the reviewers have been addressed in a satisfactory manner. As such, I am happy to recommend the acceptance of your manuscript for publication.

Congratulations on your excellent work, and thank you for your efforts in improving the paper.

Best regards,

Comments from PLOS Editorial Office: We note that one or more reviewers has recommended that you cite specific previously published works in an earlier round of revision. As always, we recommend that you please review and evaluate the requested works to determine whether they are relevant and should be cited. It is not a requirement to cite these works and you may remove them before the manuscript proceeds to publication. We appreciate your attention to this request.

---

## [Editor Report · Acceptance letter]

2 Jan 2025

PONE-D-24-06667R3 

PLOS ONE

Dear Dr. Azadi, 

I'm pleased to inform you that your manuscript has been deemed suitable for publication in PLOS ONE. Congratulations! Your manuscript is now being handed over to our production team.

Kind regards, 

on behalf of

Dr. Jose Balsa-Barreiro 

Academic Editor

PLOS ONE